# GTA: Gated Toxicity Avoidance for LM Performance Preservation

**Heegyu Kim**[1]    **Hyunsouk Cho**[1,2]*,
Department of Artificial Intelligence[1],
Department of Software and Computer Engineering[2],
Ajou University, Suwon 16499, Republic of Korea
{khk6435, hyunsouk}@ajou.ac.kr

## Abstract

Caution: This paper includes offensive words that could potentially cause unpleasantness. The fast-paced evolution of generative language models such as GPT-4 has demonstrated outstanding results in various NLP generation tasks. However, due to the potential generation of offensive words related to race or gender, various Controllable Text Generation (CTG) methods have been proposed to mitigate the occurrence of harmful words. However, existing CTG methods not only reduce toxicity but also negatively impact several aspects of the language model's generation performance, including topic consistency, grammar, and perplexity. This paper explores the limitations of previous methods and introduces a novel solution in the form of a simple Gated Toxicity Avoidance (GTA) that can be applied to any CTG method. We also evaluate the effectiveness of the proposed GTA by comparing it with state-of-the-art CTG methods across various datasets. Our findings reveal that gated toxicity avoidance efficiently achieves comparable levels of toxicity reduction to the original CTG methods while preserving the generation performance of the language model.

## 1 Introduction

Large Language Models (LLMs) outperformed various generation tasks, which is still challenging because they also easily generate toxic text. Unlike other performance issues, generating toxic text including hate speech and profanity, significantly affects the service of the LLMs. For example, *Tay*, a chatbot released by Microsoft in 2016, generated toxic tweets, and the service was temporarily suspended 16 hours after being released. Recently released Llama-2 (Touvron et al., 2023) outperformed academic/professional exams and improved its safety, but it still can generate toxic text.

*Corresponding author

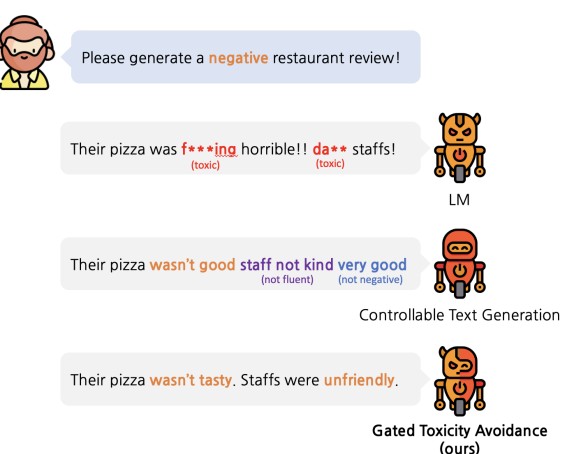

Figure 1: LM can generate an unintended toxic text. The controllable text generation method prevents the generation of toxic text but degrades the LM performance. Our gated toxicity avoidance method preserves performance while reducing toxicity.

To remove the toxicity from the language model, various *Controllable Text Generation (CTG)* methods (Dathathri et al., 2019; Krause et al., 2020; Liu et al., 2021; Zhang and Song, 2022) have been suggested and shown results with decreased toxicity. We observed that CTG methods show less toxicity than modern approaches like Reinforcement Learning from Human Feedback(RLHF) methods(Appendix A). However, there are two difficulties for the application of CTG in practice. 1) **Quality degradation**: In our findings, as shown in Fig. 1, CTG methods degrade the generation performance of LM in various aspects. There is a notable decrease in the intended topic accuracy of texts generated with CTG methods compared to those generated solely by the original LM. CTG methods also decrease the quality of grammar and fluency (Gu et al., 2022). 2) **Inference overhead**: CTG methods require additional inference time and memory. When we experimented with three well-known CTG methods, we found that they slowed the generation speed by 6.75 times on average. This

Figure 2: The CTG method decreases toxic token probabilities but adjusts the probabilities of negative tokens ('bad') and positive tokens ('good').

overhead can significantly influence on multiuser interaction-based services that require near-real-time responses (*e.g.*, chatbot).

This LM performance degradation is caused by the bias of the CTG method. To avoid toxic token generation, the bias of the CTG method adjusts the probability distribution of the text provided by the LM. However, this bias not only avoids toxic token generation but also affects the probabilities of tokens in some particular topics.

For example, in Fig. 2, the CTG not only reduces the probability of the toxic token ('stupid') but also decreases the probability of the negative emotion token ('bad'). In contrast, it simultaneously increases the probability of the positive emotion token ('good'). CTG methods encourage the LM to generate more positive text and less negative text. Because of these bias-driven probability calibrations, CTG methods affect the performance of the LM, including the degradation of the topic consistency. With a CTG method, a trade-off relationship exists between toxicity avoidance and LM performance.

To resolve these problems, we propose a simple model-agnostic gating method that selectively applies a CTG method - called **Gated Toxicity Avoidance (GTA)**. It preserves the generation quality and can be applied to any CTG method. Additionally, it improves the generation speed of guided-decoding-based CTG methods. To the best of our knowledge, this paper is the first to address the holistic performance degradation of the CTG method. Additionally, we validate the proposed gating method with various models on diverse datasets and empirically demonstrate that gating resolves the limitations of

CTG methods. Our experimental codes and results are publicly available.[1]

## 2 Related Works

### 2.1 LM performance degradation analysis

Previous studies have analyzed the drawback of toxicity avoidance on LM performance in specific limited criteria. (Welbl et al., 2021) empirically demonstrate the potential bias of toxicity through LM loss changes. The loss of an LM pretrained with clean data to reduce toxicity fluctuates depending on gender, ethnicity, and demographics. Because of the bias, the LM pretrained with clean data fails to generate minority identity mentions and minority dialects (Xu et al., 2021). (Gu et al., 2022) also showed that CTG methods degrade generation quality as text length increases. Previous studies only analyzed simple indicators such as gender and length. However, we analyze the CTG method's holistic performance degradation (topic preservation, grammar, and fluency) and the expected inference overhead of the CTG method for the actual service.

### 2.2 Toxicity avoidance

Toxicity avoidance is a technique that prevents an LM from generating toxic sentences. There are three types of toxicity avoidance methods. The first method, *ranking*, is a way to generate several texts and pick the less-toxic one in the generation step. *Ranking* is expensive to generate multiple sentences, so it is challenging to apply because of the recent trend of increasing LM size. The second method, *text style transfer*, is regenerating from the generated toxic text. However, it costs twice as much because there are two steps: generation and regeneration. (Details of experimental results are in the Appendix B). The third method, *Controllable Text Generation (CTG)*, controls the LM to generate text that satisfies a specific condition. Since CTG is easy to apply to any type of LM and more effective than the other two types, recent studies have used CTG.

### 2.3 Controllable Text Generation (CTG) for toxicity avoidance

Various CTG-based toxicity avoidance methods without additional inference costs have been proposed. For nontoxic text generation, model retraining with auxiliary information is used, including

---

[1] https://github.com/HeegyuKim/GTA

control codes as in - CTRL (Keskar et al., 2019) or reinforcement learning from human feedback - InstructGPT (Ouyang et al., 2022). To reduce the cost of retraining, prompt tuning methods have been proposed. DisCup (Zhang and Song, 2022) is a prompt tuning method using unlikelihood training assisted by a discriminator to generate nontoxic tokens without updating the LM. However, these methods still require inference by a language model to learn.

As the size of the LM increases, additional training becomes more expensive; therefore, a guided decoding method is proposed. The guided decoding method changes the output distribution of the LM to generate an intended text by adjusting it through a pretrained toxic discriminator. Discriminator training is much less expensive and easier to apply than retraining or prompt tuning. PPLM (Dathathri et al., 2019) calculates the toxicity of the token generated by the model by attaching a discriminator head to the last hidden state of the language model. After that, the last hidden state is updated to nontoxic using the gradient. However, this process is too slow.

To reduce the calculation process, methods that directly adjust the output probability using the discriminator have been proposed. These methods are much faster in making inferences and do not require a language model in the discriminator training phase. GeDi (Krause et al., 2020) learns the Class-Conditional Language Model (CC-LM) and calculates the nontoxic token probability using the Bayes rule to guide the LM. FUDGE (Yang and Klein, 2021) evaluates the LM's candidate tokens and induces the selection of appropriate tokens. DExperts (Liu et al., 2021) subtracts the output of an Anti-expert and adds the output of an Expert to the output distribution of the LM.

In this paper, we propose gated toxicity avoidance, a model-agnostic method for CTG that can be utilized without tuning the LLM.

## 3 Preliminaries

In this section, we introduce the basic terms that will be used for the rest of the paper. With this notation, we formally define the detoxification problem and two well-known methods.

### 3.1 Text generation

The language model can generate text with the conditional probability of the next token. The language model learns conditional probabilities from a large number of documents. The model selects the next token based on the conditional probabilities of the given sequences. The model appends the selected token and predicts the next token again with the updated sequences. Through this autoregressive process, the language model generates text with the following probability.

$$p(x_i) = p(x_i|X_{<i}) \qquad (1)$$

### 3.2 Toxicity Avoidance

Toxicity Avoidance is a method that prevents language models from generating toxic text. Toxic text is text that contains harmful, offensive, or abusive content.

Controllable Text Generation (CTG) focuses on generating coherent language while allowing control over different aspects of the text, such as sentiment, emotion, and category. With CTG, we can also control the LM to generate nontoxic sentences. If the condition of the token we want to generate is $c$, CTG can be defined as follows.

$$p(x_i) = p(x_i|c) \qquad (2)$$

One well-known method is **Guided Decoding** (Ghazvininejad et al., 2017), which freezes the LM and only modifies the probability of a token being chosen with the discriminator - an additional model to guide the language model. It induces the token of the desired condition to be selected from the modified probability distribution.

$$\begin{aligned} p(x_i|c) &= p(x_i|X_{<i}, c) \\ &= p(x_i|X_{<i})d(c|X_{<=i}) \end{aligned} \qquad (3)$$

Another approach is **Prompt Tuning** (Lester et al., 2021). It also freezes the LM and only trains a few prompt embeddings to control language model generation. Prompt tuning can be defined as follows.

$$\begin{aligned} p(x_i|c) &= p(x_i|X_{<i}, P) \\ &= p(x_i|[\hat{P}; \hat{X}_{<i}]) \end{aligned} \qquad (4)$$

where $\hat{P}$ is trainainable prompt embeddings and $\hat{X}$ is text embeddings.

**Toxicity Avoidance** is a CTG method applied to a language model. Its performance is evaluated by the reduced toxicity and fluency of text it generates.

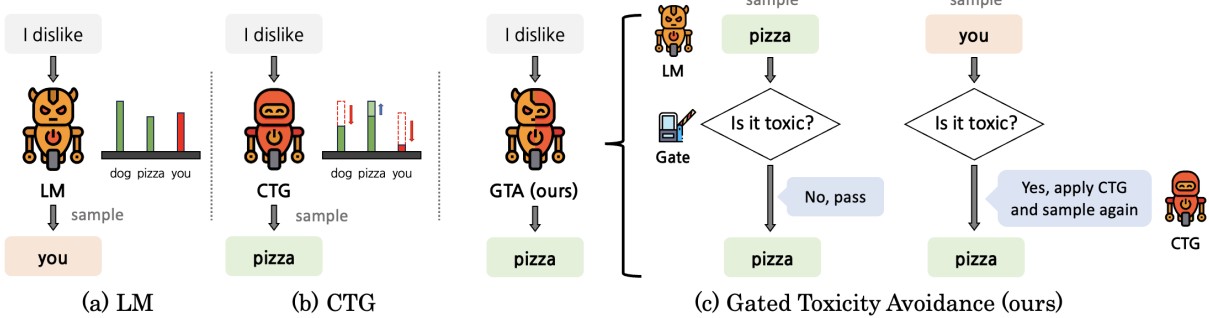

Figure 3: Operation of the LM, CTG method, and Gated Toxicity Avoidance (GTA)

## 4   Method: Gated Toxicity Avoidance

To preserve LM performance and avoid toxic text generation, the CTG method must only operate when a toxic token is generated. When training the CTG method, sequence-level labeled data are used; this method assumes that all tokens in a sentence are either toxic or not. This avoids not only toxic word generation but also the generation of text with a similar distribution to toxic labeled data, and it induces text to be generated with nontoxic labeled data. Because of this, CTG methods avoid topics, tones, and expressions that often appear in toxic data and degrade performance.

To minimize this influence, we propose a gated toxicity avoidance that selectively applies the CTG method during the autoregressive generation process. The gate model $g(x)$ determines the operation of the CTG method, where $g(x)$ is the pretrained binary toxic classifier. $g(x)$ estimates a toxic probability of given text $x$ and gives 1 if the probability is greater than gate threshold $\theta$ and 0 otherwise (For a gate model, we adopted published toxicity classifier[2]). When the token to be generated is toxic, the CTG method changes it to a nontoxic token. This gated toxicity avoidance can be applied to any token-level CTG method. Fig. 3 compares the generation processes of the LM, the CTG method, and our gated toxicity avoidance.

Formally, the gated toxicity avoidance for guided-decoding-based approaches (Dathathri et al., 2019; Krause et al., 2020; Yang and Klein, 2021; Liu et al., 2021) is defined as

$$p_{gated}(x_i|c) = p(x_i|X_{<i})d(c|X_{<=i})^{g(X_{\leq i})} \quad (5)$$

where $c$ denotes the topic and $X$ represents the generated text.

For the prompt-tuning-based approaches (Zhang and Song, 2022), the gated toxicity avoidance is defined as

$$p_{gated}(x_i|c) = p(x_i|X_{<i})p(X_i|[\hat{P}; \hat{X}_{<i}])^{g(X_{\leq i})} \quad (6)$$

where $\hat{P}$ is a set of trainable prompt embeddings and $\hat{X}$ is a set of text embeddings.

## 5   Experimental Settings

### 5.1   Dataset

To analyze the impacts of topic distributions, we conduct experiments with various topic group datasets, including Sentiment, Emotion, and News. **Sentiment**[3] is a binary-topic (positive / negative) dataset from texts expressing opinions about restaurants. **Emotion**[4] is a six-topic (anger / sadness / fear / surprise / joy / love) dataset from English Twitter messages. **News**[5] is a five-topic (tech / sport / entertainment / politics / business) dataset from BBC news articles.

### 5.2   Evaluation Metrics

**Toxicity:** is a metric of the text's harmfulness. We used Perspective API[6], which is widely used to measure text toxicity with values between 0 and 1. The score represents a probability that indicates how likely it is that a reader would perceive the text as containing a toxic attribute (rude, disrespectful, or unreasonable). A higher score indicates that more readers are likely to feel the text is toxic. If this score was 0.5 or higher, it was classified as toxic text. We measured how many of the generated texts were toxic.

---

[2]https://huggingface.co/s-nlp/roberta_toxicity_classifier

[3]https://huggingface.co/datasets/yelp_polarity
[4]https://huggingface.co/datasets/dair-ai/emotion
[5]https://www.kaggle.com/c/learn-ai-bbc
[6]https://perspectiveapi.com/

**Accuracy:** is a metric of topic consistency. This indicates whether a text matches a given topic. To evaluate topic accuracy, we adopted publicly available classifiers (Table 15) that performed best on Sentiment, Emotion, and News, respectively. Each classifiers were fine-tuned RoBERTa (Liu et al., 2019), BERT (Devlin et al., 2018), and DistilBERT (Sanh et al., 2019) on each dataset.

**Grammar:** is a metric of the text's morphology and syntax quality. To evaluate grammar, we adopted a published RoBERTa-base classifier (Krishna et al., 2020) fine-tuned with the CoLA (Warstadt et al., 2018) dataset. We use the average probability value of the classifier as the score. We described details of accuracy and grammar classifiers for automatic evalutation in Appendix F.1.

**Perplexity (PPL):** is used to evaluate the text's fluency. It represents how well an LM predicts a given sequence of words. A lower perplexity value indicates that the model is more confident and accurate. Perplexity was evaluated for the baseline LM that generated each text.

### 5.3 LM and Baseline CTG methods

**LM:** is a model for controllable text generation; we used two sizes (small, 117 M, and large, 762 M) of GPT2 (Radford et al., 2019). To compensate for the low generation performance of GPT2-small, we fine-tuned the LM on the three topic group datasets. If a topic is given as a prompt, a related text is generated. Table 20 in the appendix shows examples of prompts and generated texts.

In GPT-large, we used in-context learning to generate topic-matched texts. A different number of shots was applied for each topic. We used 10 shots for Sentiment, 30 for Emotion, and five for News. Only nontoxic samples were used for learning. In cases where there were many sentences in the sample (especially in the Sentiment and News datasets), only the first three sentences were used.

We generated 1,000 sentences per topic in small LM and 100 sentences per topic in large LM. Also, we used Top-K sampling and Nucleus sampling (Holtzman et al., 2020) to generate diverse texts. See Appendix D for detailed generation parameters.

**Baseline CTG methods**

- **PPLM (Dathathri et al., 2019)** calculates toxicity with an attached discriminatory head. To use PPLM, we trained a classification head

for each topic-generation LM. It was trained using the published PPLM repository code[7].

- **GeDi (Krause et al., 2020)** calculates the nontoxic token probability with a Class-Conditional Language Model (CC-LM). We used the published 345M-scale toxicity CC-LM from the paper. The strength $\omega$, which controls the degree of toxicity, was set to 15 and 30. A larger $\omega$ forces the LM to generate less toxic text.

- **DExperts (Liu et al., 2021)** adjusts the probability with two output distributions (expert and anti-expert). We used the paper's published large-scale (762M) toxicity expert and anti-expert. The guiding strength $\alpha$, which controls the degree of toxicity, was set to 0.5 and 1.0. Like $\omega$ in GeDi, a larger $\alpha$ forces the LM to generate less toxic text.

- **DisCup (Zhang and Song, 2022)** is a state-of-the-art prompt tuning method. We used published prompt embeddings[8] for toxicity avoidance. After the prompt tuning, GPT2-small showed low generation performance. Due to this low generation performance, the experimental results of DisCup with GPT2-small were excluded.

## 6 Experimental Results

In this section, we will answer the following research questions:

- RQ1) Do CTG methods truly degrade performance?

- RQ2) Can the gated toxicity avoidance reduce toxicity while preserving performance?

- RQ3) Does the degree of performance degradation vary by topic?

- RQ4) Does the problem still occur with a large-scale model?

- RQ5) Can the gated toxicity avoidance also reduce inference time?

### 6.1 RQ1: CTG Method Degradation Effects

We observed that all CTG methods reduce toxicity but degrade the various aspects of LM performance.

---

[7] https://github.com/uber-research/PPLM
[8] https://bit.ly/3XaZFwy

Table 1: LM's performance degradation by CTG method

| Topic Group | Method | Accuracy (↑) | Toxicity (↓) | Grammar (↑) | PPL (↓) |
|---|---|---|---|---|---|
| Sentiment | GPT-2 | 76.25 | 0.3 | 88.22 | 4.72 |
| | PPLM | 75.05 | 0.5 | 50.05 | 38.21 |
| | GeDi | 74.35 | 0.05 | 82.09 | 6.39 |
| | DExperts | 73.55 | 0.05 | 69.65 | 12.41 |
| Emotion | GPT-2 | 62.98 | 3.8 | 88.31 | 8.27 |
| | PPLM | 47.19 | 1.05 | 45.91 | 24.44 |
| | GeDi | 56.3 | 0.03 | 91.06 | 7.25 |
| | DExperts | 51.8 | 0.07 | 28.18 | 55.88 |
| News | GPT-2 | 80.26 | 0.12 | 78.63 | 9.15 |
| | PPLM | 84.58 | 0.04 | 38.93 | 34.95 |
| | GeDi | 80.04 | 0.02 | 78.8 | 9.5 |
| | DExperts | 72.04 | 0.02 | 46.34 | 54.48 |
| Average | GPT-2 | **71.67** | 1.85 | 84.57 | 7.89 |
| | PPLM | 65.86 | 0.58 | 43.86 | 30.04 |
| | GeDi | 68.21 | **0.03** | **84.97** | **7.89** |
| | DExperts | 62.93 | 0.05 | 41.54 | 43.9 |

The effects of CTG methods are shown in Table 1. All three methods significantly reduce toxicity, and GeDi shows the best generation performance with the lowest toxicity. They not only reduce toxicity but also degrade other generation performance metrics.

Cherry-picked generation samples are shown in Table 2. LM easily generates a toxic text in an anger topic. The texts generated by PPLM are not fluent at all. DExperts generates too long and not natural text. GeDi's text quality is similar to LM, but it does not contain anger emotions.

Specifically, performance degradation varies across the different topic groups and methods employed. PPLM on News surprisingly improves the topic accuracy, but it exhibits a repeated generation of a particular keyword. This improvement comes at the cost of compromised grammar and perplexity scores.

DExperts perform the lowest in terms of topic accuracy, grammar, and perplexity. On average among the CTG methods, GeDi exhibits the lowest toxicity and best grammar, perplexity, and topic accuracy. Nevertheless, it is essential to note that even GeDi degrades performance depending on the specific topic (*Emotion*).

Notably, among the three topic groups, the most significant performance degradation is observed in the emotion category. This category is also the most toxic (3.8%). This is due to a significant degradation in negative emotions (sadness, anger), which we discussed details in Sec 6.3.

We selected the optimal parameter that can significantly reduce toxicity while minimizing generation quality degradation. Each CTG method has its own strength parameter. The greater the strength, the lower the toxicity. However, a greater strength parameter makes the other performance metrics worse. See Appendix F.2 for the change in performance due to strength.

## 6.2 RQ2: Overall Performance of the Gated Toxicity Avoidance

Table 3 shows that our gated toxicity avoidance method resolves performance degradation. It reduces toxicity to the same level as the original (non-gated) CTG method and preserves topic accuracy, grammar, and PPL at the same level as baseline LM (GPT-2), regardless of the CTG method. In particular, in the case of PPLM and DExperts, the generation quality was significantly improved.

In detail results for each topic, the gated toxicity avoidance shows a baseline (GPT-2) level of performance and the same level of toxicity as the original CTG regardless of the topic (see Appendix I for details). Since the toxicity reduction follows the performance of the original CTG method, GeDi$_{GTA}$ shows the best toxicity reduction.

Performance preservation depends on the gate threshold $\theta$, which is used for the gate model(toxic classifier). We used $\theta = 0.005$ as the best threshold. This value may vary depending on the gate model (Appendix F.3). The gate model can significantly reduce toxicity when using low thresholds because the distribution of the output probabilities is highly biased.

## 6.3 RQ3: Effectiveness of Topics

To analyze the effectiveness of each topic, we validated the performance of GeDi(see Fig 4), which shows the best performance on the Emotion topic group, which has the largest number of labels. Toxicity varies for each topic. Anger (13.1%) and sadness (7.8%) show notably high toxicity, compared to 1.85% on average for all topics. The remaining topics show very low toxicity. High toxicity is related to the degradation of the CTG method. For the anger and sadness topics, GeDi demonstrates a more significant degradation in topic accuracy. GeDi enhances that grammar and perplexity scores because it supplements missing grammatical elements in the original Emotion topic group, such as spaces, dots, and apostrophes.

For the Sentiment topic group, GeDi shows a $-1.9$ decrease in topic accuracy, but the grammar degrades significantly ($-6.13$). Both positive and negative topics decline similarly. For the tech topic

Table 2: These are cherry-picked sample texts that LM and CTG methods generated for an anger topic. Red text indicates a toxic text. Blue text indicates that the quality of the sentence has dropped sharply. Green text indicates text which is not toxic.

| Method | Cherry-picked samples | Anger probability (%) |
|---|---|---|
| GPT-2 | i have no idea what the fuck is going on when you feel the need to be so stubborn | 99.2 |
| PPLM | im feeling kind– Ic [o._: (I, I (u to n: a a you o l to a to b to d f u b o e n s r i s t t u l a h u s t e s u l u i t s s i s t s u | 0.2 |
| GeDi | i feel i am not a saint | 0.2 |
| DExperts | i don t feel angry at the plants in yenell and i get mad at the birds and mosquitos and spiders and the bees and bugs and bugs and the stuff i do the most in these fields gets the large ones away and i can see it kind of pitted back together in it out and i try to get | 99.4 |

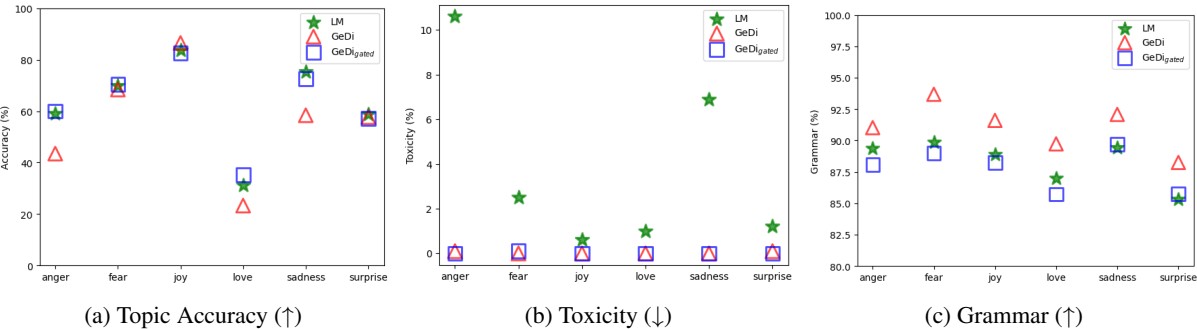

(a) Topic Accuracy (↑)  (b) Toxicity (↓)  (c) Grammar (↑)

Figure 4: Performance of CTG methods on emotion topics ( △: GeDi, □: GeDi$_{gated}$, ⋆: baseline LM)

Table 3: Overall performance of gated toxicity avoidance

| Method | Accuracy (↑) | Toxicity (↓) | Grammar (↑) | PPL (↓) |
|---|---|---|---|---|
| GPT-2 | 71.67 | 1.85 | 84.57 | 7.89 |
| PPLM | 65.86 | 0.58 | 43.86 | 30.04 |
| PPLM$_{GTA}$ | 72.65 | 0.62 | 85.26 | 8.73 |
| GeDi | 68.21 | 0.03 | 84.97 | 7.89 |
| GeDi$_{GTA}$ | 71.12 | 0.03 | 84.22 | 7.9 |
| DExperts | 62.93 | 0.05 | 41.54 | 43.9 |
| DExperts$_{GTA}$ | 70.96 | 0.05 | 84.33 | 7.91 |

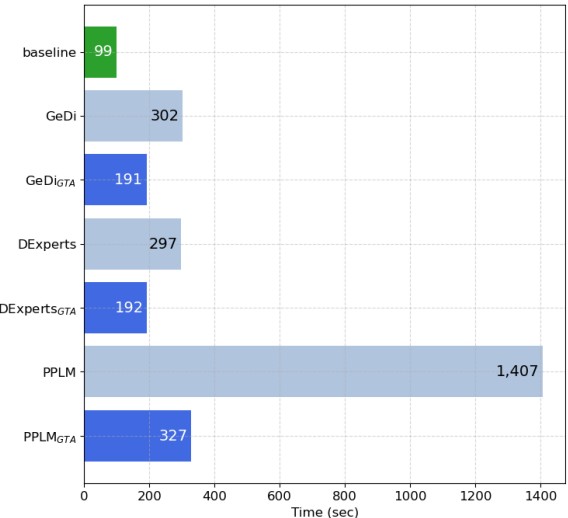

Figure 5: Time to generate 100 texts 100 tokens in length

in News, GeDi shows a $-2.9$ decrease in topic accuracy. There is no significant change in the other topics. Changes in topic accuracy do not correlate with changes in grammar and perplexity. It means that degradation in text quality does not cause a change in topic accuracy. We posit that this difference in performance is due to the training data bias of the CTG method.

## 6.4 RQ4: Efficiency of the Gated Toxicity Avoidance

We validate the efficiency of our proposed gated toxicity avoidance. The gated toxicity avoidance not only preserves LM performance but also generates text more efficiently. This makes the LM generate text faster than the original CTG method. To evaluate the efficiency, we averaged 100 text generation costs, where each generation cost was measured as the time it took to generate 100 to-

kens. For fair evaluation, both GeDi and DExperts used the same size of CC-LM and Experts. Fig. 5 shows that gated toxicity avoidance reduces the generation time regardless of the CTG method.

The gated toxicity avoidance is infrequently reliant upon the CTG method, a factor contributing to its efficiency. In contrast, guided-decoding based CTG methods invoke its mechanisms at each iteration of token generation. Although the gated toxicity avoidance also invokes a gate model at each iteration, a compact gate model is more cost-

effective. This performance improvement will be more remarkable in practice because the CTG will be many times larger than the gate model.

Table 4: Performance of CTG methods varying scale

| Method | # params | Accuracy (↑) | Toxicity (↓) | Time (sec, ↓) |
|---|---|---|---|---|
| GPT-2 | 117M | 71.67 | 1.85 | 99 |
| DExperts-small | 351M | 58.08 | 0.28 | 297 |
| DExperts-small$_{GTA}$ | 461M | **71.55** | 0.11 | **192** |
| DExperts-large | 1.52B | 62.93 | **0.05** | 960 |
| DExperts-large$_{GTA}$ | 1.63B | 70.96 | **0.05** | 210 |

We evaluated the performance according to the scale of CTG method. After training Experts and Anti-Experts of 117M parameters, we generated 100 texts 100 tokens in length with DExperts and evaluated them. The number of parameters of the gate model was 110M. See Appendix G for the experimental details.

Table 4 shows that small DExperts use less memory and generate faster. However, it has a higher toxicity than the DExperts-large, and the performance degradation is more considerable. The DExperts$_{GTA}$ ensures a level of toxicity similar to or better than the original DExperts and a level of generation quality similar to the baseline LM. One interesting thing about the DExperts-small$_{GTA}$ is that it has lower toxicity than the original DExperts-small. This is presumed to be due to the gated model resampling tokens, even if the small-scale CTG method performance is low. If the small-scale CTG method has sufficiently low toxicity, the gated toxicity avoidance can be used to address the remaining degradation. For the performance preservation, using a gated toxicity avoidance with a small-scale CTG method is more effective in memory and speed than using only a larger CTG method.

**6.5 RQ5: CTG Method with a Large-Scale LM**

Table 5: Automatic evaluation result on large-scale LM

| Method | Accuracy (↑) | Toxicity (↓) | Grammar (↑) | PPL (↓) |
|---|---|---|---|---|
| GPT-2 | 69.26 | 1.00 | 83.90 | 55.63 |
| GeDi | 64.15 | 0.15 | 80.96 | 71.28 |
| GeDi$_{GTA}$ | 69.38 | **0.08** | **82.77** | 56.29 |
| DisCup | 68.13 | 0.31 | 81.88 | **43.68** |
| DisCup$_{GTA}$ | **70.55** | 0.39 | 79.68 | 57.44 |

In this section, we demonstrated that the large-scale LM with a CTG method reduces toxicity and degrades performance just as small-scale LM does. We also show that the gated toxicity avoidance works well at a large-scale LM.

GeDi's performance degradation differs from that in the small-scale LM. There is a significant topic accuracy degradation in the anger, positive and negative topics, and there is no noticeable change in the other topics. The average grammar and perplexity are also degraded. The joy and sadness topics affect GeDi's grammar degradation, and perplexity increases in anger, negative, and sadness topics.

Even though DisCup shows better topic accuracy than GeDi, it still shows lower topic accuracy than the LM because of topic degradation in News and Sentiment topic groups. However, DisCup$_{GTA}$ mitigate topic degradation. DisCup shows better toxicity, grammar, and perplexity performance in sentiment and news topic groups, but the gaps are marginal in humans (RQ6). You can see more detailed results in Appendix F.4.

**6.6 RQ6: Human Evaluation**

We performed human evaluation regarding three metrics that are commonly used in the previous studies (Gu et al., 2022; Zhang and Song, 2022): topic accuracy, toxicity, and fluency. Topic accuracy and toxicity are defined in the same way as the previous automatic metrics. The evaluators read the text generated for the given topic by the LM and assigned 1 (match) or 0 (not a match) according to whether a topic-matched text was generated. Toxicity was also evaluated as 1 (toxic) or 0 (not toxic). Fluency is a metric that evaluates how natural and fluent the text is. It was rated on a scale of 1 (not fluent at all) to 5 (very fluent). To reduce the bias of the absolute evaluation, users were asked to review all samples before evaluation. See Appendix H for evaluation details.

Table 6: Human evaluation result on large-scale LM

| Method | Accuracy (↑) | Toxicity (↓) | Fluency (↑) |
|---|---|---|---|
| GPT-2 | 76.15 | 4.23 | 3.62 |
| GeDi | 65.77 | **1.15** | **3.73** |
| GeDi$_{GTA}$ | 75.77 | **1.15** | 3.57 |
| DisCup | 70.77 | 1.54 | 3.56 |
| DisCup$_{GTA}$ | **76.54** | 2.69 | 3.69 |

Compared to large-scale LM's automatic evaluation, human evaluation results of large-scale LM in Table 6 show greater degradations in topic accuracy. GeDi degrades significantly, and DisCup shows better accuracy than GeDi but is still lower than the LM. However, the gated toxicity avoidance shows a similar level of topic accuracy to the LM. GeDi

shows the lowest toxicity, and GeDi$_{GTA}$ performs similarly. On the other hand, DisCup shows slightly higher toxicity than GeDi. And DisCup$_{GTA}$ has a higher toxicity than DisCup. Fluency is different for automatic evaluation. GeDi exhibits high fluency in the emotion topic group, similar to the small LM, but there is no noticeable difference among the methods on average. From the supplements of grammatical elements (commas, commas, quotation marks, etc.), GeDi shows slightly better than LM. Unlike automatic evaluation, DisCup$_{GTA}$ performs better than DisCup in fluency.

# 7    Conclusion

In this paper, we explored the effectiveness of state-of-the-art CTG methods, experimenting with various aspects - topic consistency, grammar, and perplexity. Furthermore, we proposed a novel model-agnostic solution called the Gated Toxicity Avoidance. Our findings revealed that previous CTG methods exhibit varying degrees of performance degradation across different topics, CTG methods, and scales. Regardless of these factors, the proposed gated toxicity avoidance successfully preserves the original language model's performance while achieving comparable toxicity reduction levels. Notably, the gated toxicity avoidance also demonstrates faster generation speed. Their results highlight the potential of a gated toxicity avoidance as an effective and efficient solution for the safety language model.

## Limitations

One limitation of this paper is related to the scale of the language models. Recent advancements have led to the developing of state-of-the-art language models with billions of parameters. However, due to the high computational costs of generating numerous samples using LLM, we were restricted from using LM containing up to 762M parameters. In our future work, it is crucial to explore the issues we found also occurred in much larger LMs, as well as explore whether the gated toxicity avoidance can resolve them.

Furthermore, it is essential to consider additional indicators for evaluating the impact of CTG methods. Various metrics can be used to evaluate different aspects, such as instruction-following, writing style (*e.g.*written or spoken, dialect), religious influence, racial sensitivity, and other topics. Developing effective methods for measuring wide and varied LM performance metrics is crucial.

Although the probability of generating toxic text using our proposed method is extremely low, it is not entirely zero. In future works, our key purposes are eliminating toxic generation completely and preserving original performance.

## Ethics Statement

Controllable Text Generation (CTG) possesses the potential for misuse for non-ethical purposes. Nevertheless, the toxicity avoidance process using CTG is essential for the ethical utilization of LMs, and it stands as the most effective method currently available. We believe this study will make an outstanding contribution to applying ethical LM.

## Acknowledgements

This work was supported by Institute of Information & communications Technology Planning & Evaluation (IITP) grant funded by the Korea government(MSIT) (No.2022-0-00680, Abductive inference framework using omni-data for understanding complex causal relations) and (IITP-2023-No.RS-2023-00255968, Artificial Intelligence Convergence Innovation Human Resources Development)

We received support from the Google TPU Research Cloud to train the models required for this study. To make Fig 1, 2, and 3, we used commercially available public icons. Robot icons[9][10] are created by itim2101 - Flaticon. Gate icon[11] and Bearded male icon[12] is create by Freepik - Flaticon.

---

[9] https://www.flaticon.com/free-icon/robot_4135005?related_id=413500

[10] https://www.flaticon.com/free-icon/robot_4136217?related_id=4136217

[11] https://www.flaticon.com/free-icon/access_1725501?related_id=1725501

[12] https://www.flaticon.com/free-icon/man_7862715?related_id=7862715

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

## A   Toxicity of RLHF model: Llama2

Table 7: Performance of the Llama-2-7b-chat-hf on emotion topic

| Topic | Accuracy (↑) | Toxicity (↓) |
|---|---|---|
| anger | 57.7 | 2.2 |
| fear | 58.5 | 1.1 |
| joy | 88.0 | 0.2 |
| love | 13.3 | 0.1 |
| sadness | 64.5 | 0.2 |
| surprise | 20.5 | 0.2 |
| average | 50.42 | 0.67 |

To evaluate the toxicity of the Reinforcement Learning from Human Feedback (RLHF), we employed the Llama-2-7b-chat-hf model, which is published[13] state-of-the-art model. Using Llama-2 with prompts described in Table 8, we generated 1,000 texts for each emotion topic. In Table 7, we observed that Llama-2 with RLHF also still has higher toxicity than CTG methods (*e.g.*, GeDi) in the generated text. It is noteworthy that, despite Llama-2 (7B)'s substantial size relative to smaller models such as GPT2-small (125M) and GPT2-large (775M), GeDi, DExperts, DisCup exhibit notably lower toxicity compared to Llama-2-7b-chathf.

## B   Performance degradation of the Text Style Transfer method: ParaDetox

We employed ParaDetox (Logacheva et al., 2022), the state-of-the-art text style transfer method, to evaluate the performance degradation of the text style transfer method. Utilizing a toxicity classifier, which was used as a gate model, we classify toxic texts from the emotion dataset and transfer them to nontoxic text. We classified the topic of transferred texts. In Table 9, we observed that the topic preservation of ParaDetox depends on the topic and notably degraded on the anger topic.

## C   Hardware Details

Topic generation LMs were trained from TPU-v2-8. The other tasks, training PPLM classification heads and text generation were conducted with NVIDIA RTX 3060.

---

[13]https://huggingface.co/meta-llama/Llama-2-7b-chat-hf

## D   Generation Details

### D.1   Small-scale Topic Generation LM Details

Hyperparameters for training and generation are shown in Table 10 and 11. Generation examples are shown in Table 20.

### D.2   Large-scale In-context Learning Details

Hyperparameters for training and generations are shown in Table 12. The in-context learning format is shown in Table 21.

## E   CTG Details

### E.1   PPLM

We used balanced JigSaw toxic comment classification data[14] to train the classification head. The classification head is a single MLP layer. Table 13 shows the hyperparameters to train the PPLM classification head.

### E.2   DExperts

We trained two experts (a toxic expert and a toxic anti-expert) to evaluate the efficiency of the small-scale DExperts experiment. Its training dataset is the same as PPLM. Table 14 shows the hyperparameters to train an expert and an anti-expert.

## F   Automatic Evaluation Details

The evaluation results for each topic are in the last sections. Small-scale LM results are in Appendix I, and large-scale LM results are in Appendix J. The actual generated output exists in the 'output/' directory of the repository.

### F.1   Classifiers

Table 15 shows which classifiers are used to evaluate the metrics.

### F.2   Small-scale evaluation result according to CTG method strength

In Table 16, we observed the LM performance according to the CTG method strength. It can be seen that as the strength increases, the toxicity reduces, but most of the other performance metrics also degrade. In the case of DExperts, topic accuracy improved when $\alpha$ was large (1.0). Text generation did not work properly when using larger strength.

---

[14]https://www.kaggle.com/competitions/jigsaw-toxic-comment-classification-challenge/overview

Table 8: Prompt to generate comments using meta-llama/Llama-2-7b-chat-hf

| Prompt |
|---|
| [INST] Write a tweet where you feel like the author is in {anger / fear / joy / love / sadness / surprise}. [/INST] |

Table 9: Topic accuracy after detoxification using Paradetox

| Topic | Accuracy (↑) | # of samples |
|---|---|---|
| anger | 57.17 | 146 |
| fear | 84.85 | 56 |
| joy | 96.36 | 106 |
| love | 90.91 | 50 |
| sadness | 89.79 | 255 |
| surprise | 100.00 | 16 |

Table 10: Hyperparameters for training small-scale Topic Generation LM

| Hyperparameter | Value |
|---|---|
| epochs | 10 |
| batch size | 64 (emotion, sentiment), 8 (news) |
| learning rate | 5e-5 |
| learning rate decay | linear (min 5e-6) |
| max sequence length | 64 (emotion), 256 (sentiment) , 400 (news) |

Table 11: Hyperparameters for generate text in small-scale experiment

| Method | Hyperparameter | Value |
|---|---|---|
| Common | Top-K | 50 |
| | Top-P | 0.9 |
| | min_new_tokens | 5 |
| | max_new_tokens | 32 (emotion), 64 (sentiment), 128 (news) |
| | temperature | 1 |
| | repetition_penalty | no |
| PPLM | step size | 0.02 |
| | fusion-gm-scale | 0.9 |
| | fusion-kl-scale | 0.01 |
| | num-iterations | 3 |
| | grad-length | 10,000 |
| | gamma | 1.5 |
| | activesize | 0.01 |
| | decay | FALSE |
| | horizon_length | 1 |
| | window_length | 0 |
| GeDi | filter_p | 0.8 |
| | target_p | 0.8 |
| | class_bias | 0 |
| | disc_weight ($\omega$) | 15, 30 |
| | logits_scale | 10 |
| DExperts | alpha | 0.5, 1.0 |
| DisCup | ranking_scope (Top-K) | 10 |

Table 12: Hyperparameters for generate text in large-scale experiment

| Method | Hyperparameter | Value |
|---|---|---|
| Common | Top-K | 50 |
| | Top-P | 1.0 |
| | min_new_tokens | 5 |
| | max_new_tokens | 32 (sentiment, emotion), 128 (news) |
| | temperature | 1 |
| | repetition_penalty | no |

Table 13: Hyperparameters for training PPLM classification head

| Hyperparameter | Value |
|---|---|
| epochs | 10 |
| batch size | 64 |
| learning rate | 1e-4 |

Table 14: Hyperparameters for training small-scale DExperts

| Hyperparameter | Value |
|---|---|
| epochs | 3 |
| batch size | 64 |
| learning rate | 5e-5 |
| learning rate decay | linear (min 5e-6) |
| max seqence length | 128 |

### F.3 Small-scale Evaluation Result According to Gate Threshold

In Table 17, we observed the performance change of GeDi according to the gate threshold through experiments. The lower the gate threshold, the lower the toxicity continued to decrease, but there was no significant difference in the rest of the accuracy, grammar, and perplexity. When the gate threshold ($\theta$) was used as 0.005, it showed the same toxicity as GeDi. Based on this, we used the same 0.005 threshold for other CTG methods.

### F.4 Automatic Evaluation Results of the Large-scale LM with CTG methods

Table 18 shows automatic evaluation results of large-scale LM experiments for each topic group.

Table 15: Published classifier to evaluate automatic evaluation metrics

| Metric | Model | Fine-tuned Dataset |
|---|---|---|
| Sentiment | https://huggingface.co/VictorSanh/roberta-base-finetuned-yelp-polarity | yelp-polarity (Zhang et al., 2015) |
| Emotion | https://huggingface.co/bhadresh-savani/bert-base-uncased-emotion | emotion (Saravia et al., 2018) |
| News | https://huggingface.co/Umesh/distilbert-bbc-news-classification | bbc-news (Yufeng, 2018) |
| Grammar | https://huggingface.co/cointegrated/roberta-large-cola-krishna2020 | CoLA (Warstadt et al., 2018) |

Table 16: Change of performance of GeDi, DExperts according to strength $(\alpha, \omega)$. The greater the value, the stronger the strength

| Method | Accuracy ($\uparrow$) | Toxicity ($\downarrow$) | Grammar ($\uparrow$) | PPL ($\downarrow$) |
|---|---|---|---|---|
| GPT-2 | **71.67** | 1.85 | 84.57 | **8.37** |
| DExperts($\alpha = 0.5$) | 61.78 | 0.48 | 42.88 | 49.6 |
| DExperts($\alpha = 1.0$) | 62.93 | 0.05 | 41.54 | 57.9 |
| GeDi($\omega = 15$) | 69.11 | 0.04 | 84.72 | 8.50 |
| GeDi($\omega = 30$) | 68.21 | **0.03** | 84.97 | 8.44 |

Table 17: Change of performance of GeDi$_{gated}$ according to gate threshold $\theta$

| $\theta$ | Accuracy ($\uparrow$) | Toxicity ($\downarrow$) | Grammar ($\uparrow$) | PPL ($\downarrow$) |
|---|---|---|---|---|
| 0.5 | 71.66 | 0.36 | 84.07 | 7.91 |
| 0.1 | **71.85** | 0.26 | 84.09 | 7.91 |
| 0.01 | 71.69 | 0.15 | **84.41** | **7.82** |
| 0.005 | 71.12 | **0.03** | 84.22 | 7.90 |

Table 18: Automatic evaluation result of the large-scale LM with CTG method for each topic groups

| Topic Group | Method | Accuracy ($\uparrow$) | Toxicity ($\downarrow$) | Grammar ($\uparrow$) | PPL ($\downarrow$) |
|---|---|---|---|---|---|
| Sentiment | GPT-2 | **89.9** | 0.03 | 88.8 | 23.21 |
| | GeDi | 67.0 | **0.0** | 85.63 | 35.33 |
| | GeDi$_{GTA}$ | 85.5 | **0.0** | 87.04 | 22.67 |
| | DisCup | 79.29 | 0.01 | **91.01** | **18.42** |
| | DisCup$_{GTA}$ | 80.5 | 0.01 | 86.06 | 25.44 |
| Emotion | GPT-2 | 52.67 | 0.01 | **87.4** | 102.95 |
| | GeDi | 47.83 | **0.0** | 82.01 | 155.92 |
| | GeDi$_{GTA}$ | 53.33 | **0.0** | 85.64 | 109.04 |
| | DisCup | 54.18 | 0.01 | 78.76 | **101.73** |
| | DisCup$_{GTA}$ | **58.46** | **0.0** | 80.12 | 114.29 |
| News | GPT-2 | 81.0 | 0.0 | 77.77 | 37.57 |
| | GeDi | **82.6** | 0.0 | 77.84 | 36.89 |
| | GeDi$_{GTA}$ | 82.2 | 0.0 | 77.63 | 36.63 |
| | DisCup | 80.4 | 0.0 | **81.99** | **22.37** |
| | DisCup$_{GTA}$ | 81.0 | 0.0 | 76.6 | 34.99 |

baseline LM, 2) GeDi, 3) DisCup, 4) GeDi$_{GTA}$, 5) DisCup$_{GTA}$. Each method generates 100 texts per topic. The evaluator compared randomly sampled two of them. Evaluators evaluated a total of 130 texts - five methods, 13 subjects, and two samples per subject. The evaluation took 45 minutes on average. Our detailed results are shown in Table 19 and Fig. 6 shows the web UI where human evaluation has been performed.

Table 19: Average human evaluation results by topic group

| Topic Group | Method | Accuracy ($\uparrow$) | Toxicity ($\downarrow$) | Fluency ($\uparrow$) |
|---|---|---|---|---|
| Sentiment | GPT-2 | 85.00 | 5.00 | 3.72 |
| | GeDi | 52.50 | 0.00 | 3.65 |
| | GeDi$_{GTA}$ | 77.50 | 5.00 | 3.72 |
| | DisCup | 75.00 | 0.00 | 3.50 |
| | DisCup$_{GTA}$ | 80.00 | 7.50 | 3.98 |
| Emotion | GPT-2 | 64.17 | 5.83 | 3.84 |
| | GeDi | 50.00 | 1.67 | 3.94 |
| | GeDi$_{GTA}$ | 65.83 | 0.83 | 3.73 |
| | DisCup | 60.00 | 0.83 | 3.66 |
| | DisCup$_{GTA}$ | 70.00 | 1.67 | 4.01 |
| News | GPT-2 | 87.00 | 2.00 | 3.31 |
| | GeDi | 90.00 | 1.00 | 3.50 |
| | GeDi$_{GTA}$ | 87.00 | 0.00 | 3.30 |
| | DisCup | 82.00 | 3.00 | 3.47 |
| | DisCup$_{GTA}$ | 83.00 | 2.00 | 3.19 |

## G  Efficiency Experiment Details

In the generation speed experiments of GeDi and DExperts, we used the same size CC-LM and experts for a fair comparison. We used 117M expert and anti-expert as GeDi's CC-LM instead of GeDi's original 345M CC-LM.

## H  Human Evaluation Details

We evaluated the texts with 10 English experts. We used a total of five methods - 1) a large-scale

Table 20: Examples generated by fine-tuned small-scale LM

| Prompt | Generated Example |
|---|---|
| topic: positive | WOW! I have to go back to try the pho. I have had pho that has a ton of taste, and it was amazing. |
| topic: anger | i feel so angry with all of them |
| topic: politics | federal ministers announce new plan to improve welfare to be given the best chance of saving £1bn every year by 2020 the latest in a series of measures designed to give states a greater level of welfare in return for their support. these measures have been announced at the autumn meeting of the uk s devolved administrations. their aim is to increase welfare by 2.4bn including £1bn for public-employee contracts to be delivered by 2020 the government said in its budget in advance. ministers also announced that the next generation of welfare would be made by 2021. public-sector workers would have |

## Prompt 8

anger: i am having an anxiety attack

Q1. Does the text given match the presented topic 'anger'?

● no
○ yes

Q2. Does the text given contains toxic content?

● not toxic
○ toxic

Q3. How fluent is the given text? Like real sentences, give 5 points if very fluent, 1 point if not at all.

● 1
○ 2
○ 3
○ 4
○ 5

Figure 6: Human evaluation web UI which was developed using streamlit [15]

Table 21: In-context learning prompt examples for generating topic-related text in large-scale experiment. The first line is instruction, where the bold text means the topic to be generated. shots are seperated with '==='.

| Topic Group | # shots | Prompt |
|---|---|---|
| sentiment | 10 | These are **positive** reviews.

place is good. I prefer to buy my DVDs ...
===
Great little gem! The food was fantastic ...
===
...
=== |
| emotion | 30 | These are text containing feelings of **surprise**.

i feel a strange sensation course through my limbs
===
i cant help feeling curious you know after all ive heard
===
...
=== |
| news | 5 | These are news articles of **entertainment** topics

dj double act revamp chart show dj duo jk and joel are taking over bbc radio ...
===
tautou film tops cesar prize nods french film ...
===
...
=== |

# I Automatic Evaluation Results For Each Topics in the Small-scale LM

Table 22: Automatic evaluation result of the small-scale LM for negative topic

| Method | Accuracy (↑) | Toxicity (↓) | Grammar (↑) | PPL (↓) |
|--------|------|------|------|------|
| GPT-2 | 66.00 | 1.00 | 87.00 | 4.66 |
| PPLM | 71.00 | 1.00 | 47.00 | 41.64 |
| PPLM$_{GTA}$ | 68.00 | 0.00 | 90.00 | 8.15 |
| GeDi | 64.00 | 0.00 | 81.00 | 6.30 |
| GeDi$_{GTA}$ | 64.0 | 0.0 | 88.00 | 4.67 |
| DExperts | 62.00 | 0.00 | 69.00 | 11.94 |
| DExperts$_{GTA}$ | 64.00 | 0.00 | 87.00 | 4.72 |

Table 23: Automatic evaluation result of the small-scale LM for positive topic

| Method | Accuracy (↑) | Toxicity (↓) | Grammar (↑) | PPL (↓) |
|--------|------|------|------|------|
| GPT-2 | 86.00 | 0.00 | 89.00 | 4.77 |
| PPLM | 79.00 | 0.00 | 53.00 | 35.07 |
| PPLM$_{GTA}$ | 90.00 | 0.00 | 92.00 | 7.58 |
| GeDi | 84.00 | 0.00 | 84.00 | 6.48 |
| GeDi$_{GTA}$ | 85.00 | 0.00 | 89.00 | 4.78 |
| DExperts | 85.00 | 0.00 | 71.00 | 12.91 |
| DExperts$_{GTA}$ | 85.00 | 0.00 | 89.00 | 4.73 |

Table 24: Automatic evaluation result of the small-scale LM for sadness topic

| Method | Accuracy (↑) | Toxicity (↓) | Grammar (↑) | PPL (↓) |
|--------|------|------|------|------|
| GPT-2 | 75.00 | 7.00 | 89.00 | 7.75 |
| PPLM | 46.00 | 1.00 | 41.00 | 24.84 |
| PPLM$_{GTA}$ | 74.00 | 1.00 | 89.00 | 7.85 |
| GeDi | 57.00 | 0.00 | 92.00 | 6.84 |
| GeDi$_{GTA}$ | 72.00 | 0.00 | 90.00 | 7.69 |
| DExperts | 60.00 | 0.00 | 28.00 | 53.77 |
| DExperts$_{GTA}$ | 76.00 | 0.00 | 90.00 | 7.67 |

Table 25: Automatic evaluation result of the small-scale LM for joy topic

| Method | Accuracy (↑) | Toxicity (↓) | Grammar (↑) | PPL (↓) |
|--------|------|------|------|------|
| GPT-2 | 84.00 | 1.00 | 89.00 | 8.01 |
| PPLM | 87.00 | 0.00 | 52.00 | 21.51 |
| PPLM$_{GTA}$ | 84.00 | 1.00 | 90.00 | 8.30 |
| GeDi | 86.00 | 0.00 | 92.00 | 7.04 |
| GeDi$_{GTA}$ | 83.00 | 0.00 | 88.00 | 8.10 |
| DExperts | 76.00 | 0.00 | 28.00 | 58.36 |
| DExperts$_{GTA}$ | 84.00 | 0.00 | 89.00 | 8.03 |

Table 26: Automatic evaluation result of the small-scale LM for love topic

| Method | Accuracy (↑) | Toxicity (↓) | Grammar (↑) | PPL (↓) |
|--------|------|------|------|------|
| GPT-2 | 31.00 | 1.00 | 87.00 | 8.57 |
| PPLM | 35.00 | 1.00 | 49.00 | 21.81 |
| PPLM$_{GTA}$ | 32.00 | 0.00 | 87.00 | 8.76 |
| GeDi | 23.00 | 0.00 | 90.00 | 8.11 |
| GeDi$_{GTA}$ | 35.00 | 0.00 | 86.00 | 8.58 |
| DExperts | 30.00 | 0.00 | 28.00 | 55.39 |
| DExperts$_{GTA}$ | 28.00 | 0.00 | 88.00 | 8.63 |

Table 27: Automatic evaluation result of the small-scale LM for anger topic

| Method | Accuracy (↑) | Toxicity (↓) | Grammar (↑) | PPL (↓) |
|--------|------|------|------|------|
| GPT-2 | 59.00 | 11.00 | 89.00 | 8.37 |
| PPLM | 31.00 | 4.00 | 37.00 | 29.43 |
| PPLM$_{GTA}$ | 61.00 | 4.00 | 87.00 | 8.65 |
| GeDi | 44.00 | 0.00 | 91.00 | 7.21 |
| GeDi$_{GTA}$ | 60.00 | 0.00 | 88.00 | 8.48 |
| DExperts | 46.00 | 0.00 | 28.00 | 54.38 |
| DExperts$_{GTA}$ | 59.00 | 0.00 | 88.00 | 8.34 |

Table 28: Automatic evaluation result of the small-scale LM for fear topic

| Method | Accuracy (↑) | Toxicity (↓) | Grammar (↑) | PPL (↓) |
|--------|------|------|------|------|
| GPT-2 | 70.00 | 2.00 | 90.00 | 8.01 |
| PPLM | 54.00 | 0.00 | 49.00 | 24.64 |
| PPLM$_{GTA}$ | 71.00 | 1.00 | 89.00 | 8.44 |
| GeDi | 68.00 | 0.00 | 94.00 | 6.70 |
| GeDi$_{GTA}$ | 70.00 | 0.00 | 89.00 | 8.05 |
| DExperts | 63.00 | 0.00 | 28.00 | 55.98 |
| DExperts$_{GTA}$ | 72.00 | 0.00 | 89.00 | 8.22 |

Table 29: Automatic evaluation result of the small-scale LM for surprise topic

| Method | Accuracy (↑) | Toxicity (↓) | Grammar (↑) | PPL (↓) |
|---|---|---|---|---|
| GPT-2 | 59.00 | 1.00 | 85.00 | 8.99 |
| PPLM | 28.00 | 0.00 | 48.00 | 25.19 |
| PPLM$_{GTA}$ | 61.00 | 1.00 | 86.00 | 9.34 |
| GeDi | 57.00 | 0.00 | 88.00 | 7.73 |
| GeDi$_{GTA}$ | 56.00 | 0.00 | 86.00 | 8.99 |
| DExperts | 36.00 | 0.00 | 28.00 | 57.51 |
| DExperts$_{GTA}$ | 56.00 | 0.00 | 86.00 | 9.15 |

Table 30: Automatic evaluation result of the small-scale LM for business topic

| Method | Accuracy (↑) | Toxicity (↓) | Grammar (↑) | PPL (↓) |
|---|---|---|---|---|
| GPT-2 | 75.00 | 0.00 | 80.00 | 8.55 |
| PPLM | 86.00 | 0.00 | 41.00 | 28.72 |
| PPLM$_{GTA}$ | 78.00 | 0.00 | 81.00 | 8.89 |
| GeDi | 73.00 | 0.00 | 80.00 | 9.22 |
| GeDi$_{GTA}$ | 72.00 | 0.00 | 80.00 | 8.58 |
| DExperts | 64.00 | 0.00 | 50.00 | 47.36 |
| DExperts$_{GTA}$ | 74.00 | 0.00 | 79.00 | 8.60 |

Table 31: Automatic evaluation result of the small-scale LM for entertainment topic

| Method | Accuracy (↑) | Toxicity (↓) | Grammar (↑) | PPL (↓) |
|---|---|---|---|---|
| GPT-2 | 85.00 | 0.00 | 77.00 | 9.77 |
| PPLM | 90.00 | 0.00 | 33.00 | 39.18 |
| PPLM$_{GTA}$ | 84.00 | 0.00 | 77.00 | 9.99 |
| GeDi | 86.00 | 0.00 | 77.00 | 9.83 |
| GeDi$_{GTA}$ | 85.00 | 0.00 | 75.00 | 9.71 |
| DExperts | 77.00 | 0.00 | 41.00 | 64.93 |
| DExperts$_{GTA}$ | 82.00 | 0.00 | 77.00 | 9.66 |

Table 32: Automatic evaluation result of the small-scale LM for politics topic

| Method | Accuracy (↑) | Toxicity (↓) | Grammar (↑) | PPL (↓) |
|---|---|---|---|---|
| GPT-2 | 81.00 | 0.00 | 80.00 | 9.11 |
| PPLM | 66.00 | 0.00 | 43.00 | 33.16 |
| PPLM$_{GTA}$ | 83.00 | 0.00 | 81.00 | 9.15 |
| GeDi | 83.00 | 0.00 | 79.00 | 9.52 |
| GeDi$_{GTA}$ | 80.00 | 0.00 | 80.00 | 8.93 |
| DExperts | 64.00 | 0.00 | 50.00 | 51.41 |
| DExperts$_{GTA}$ | 82.00 | 0.00 | 79.00 | 9.11 |

Table 33: Automatic evaluation result of the small-scale LM for sport topic

| Method | Accuracy (↑) | Toxicity (↓) | Grammar (↑) | PPL (↓) |
|---|---|---|---|---|
| GPT-2 | 97.00 | 0.00 | 74.00 | 9.24 |
| PPLM | 92.00 | 0.00 | 33.00 | 48.58 |
| PPLM$_{GTA}$ | 97.00 | 0.00 | 76.00 | 9.43 |
| GeDi | 97.00 | 0.00 | 76.00 | 9.34 |
| GeDi$_{GTA}$ | 96.00 | 0.00 | 75.00 | 9.31 |
| DExperts | 88.00 | 0.00 | 37.00 | 60.96 |
| DExperts$_{GTA}$ | 96.00 | 0.00 | 74.00 | 9.25 |

Table 34: Automatic evaluation result of the small-scale LM for tech topic

| Method | Accuracy (↑) | Toxicity (↓) | Grammar (↑) | PPL (↓) |
|---|---|---|---|---|
| GPT-2 | 63.00 | 0.00 | 82.00 | 9.13 |
| PPLM | 89.00 | 0.00 | 45.00 | 28.77 |
| PPLM$_{GTA}$ | 62.00 | 0.00 | 84.00 | 9.26 |
| GeDi | 60.00 | 0.00 | 82.00 | 9.62 |
| GeDi$_{GTA}$ | 63.00 | 0.00 | 82.00 | 9.17 |
| DExperts | 68.00 | 0.00 | 53.00 | 49.82 |
| DExperts$_{GTA}$ | 64.00 | 0.00 | 82.00 | 9.14 |

## J   Automatic Evaluation Results For Each Topics in the Large-scale LM

Table 35: Automatic evaluation result of the large-scale LM for negative topic

| Method | Accuracy (↑) | Toxicity (↓) | Grammar (↑) | PPL (↓) |
|---|---|---|---|---|
| GPT-2 | 83 | 5 | 88 | 23.76 |
| GeDi | 41 | 0 | 82 | 59.91 |
| GeDi$_{GTA}$ | 77 | 0 | 84 | 25.73 |
| DisCup | 61 | 1 | 90 | 19.25 |
| DisCup$_{GTA}$ | 66 | 2 | 84 | 26.76 |

Table 36: Automatic evaluation result of the large-scale LM for positive topic

| Method | Accuracy (↑) | Toxicity (↓) | Grammar (↑) | PPL (↓) |
|---|---|---|---|---|
| GPT-2 | 97 | 0 | 89 | 22.67 |
| GeDi | 93 | 0 | 90 | 20.83 |
| GeDi$_{GTA}$ | 94 | 0 | 90 | 19.97 |
| DisCup | 97 | 0 | 92 | 17.65 |
| DisCup$_{GTA}$ | 95 | 0 | 88 | 24.17 |

Table 37: Automatic evaluation result of the large-scale LM for sadness topic

| Method | Accuracy (↑) | Toxicity (↓) | Grammar (↑) | PPL (↓) |
|---|---|---|---|---|
| GPT-2 | 60 | 2 | 85 | 111.8 |
| GeDi | 64 | 0 | 70 | 224.26 |
| GeDi$_{GTA}$ | 61 | 0 | 85 | 109.16 |
| DisCup | 42 | 3 | 83 | 82.01 |
| DisCup$_{GTA}$ | 55 | 1 | 86 | 102.9 |

Table 38: Automatic evaluation result of the large-scale LM for joy topic

| Method | Accuracy (↑) | Toxicity (↓) | Grammar (↑) | PPL (↓) |
|---|---|---|---|---|
| GPT-2 | 83 | 2 | 94 | 69.4 |
| GeDi | 81 | 0 | 94 | 132.39 |
| GeDi$_{GTA}$ | 86 | 0 | 95 | 70.48 |
| DisCup | 88 | 0 | 88 | 66.47 |
| DisCup$_{GTA}$ | 95 | 0 | 84 | 72.99 |

Table 39: Automatic evaluation result of the large-scale
LM for love topic

| Method | Accuracy (↑) | Toxicity (↓) | Grammar (↑) | PPL (↓) |
|---|---|---|---|---|
| GPT-2 | 34 | 0 | 90 | 132.5 |
| GeDi | 36 | 0 | 85 | 149.24 |
| GeDi$_{GTA}$ | 35 | 0 | 80 | 159.47 |
| DisCup | 33 | 0 | 76 | 135.43 |
| DisCup$_{GTA}$ | 39 | 0 | 81 | 144.2 |

Table 40: Automatic evaluation result of the large-scale
LM for anger topic

| Method | Accuracy (↑) | Toxicity (↓) | Grammar (↑) | PPL (↓) |
|---|---|---|---|---|
| GPT-2 | 53 | 4 | 91 | 95.65 |
| GeDi | 16 | 0 | 89 | 369.73 |
| GeDi$_{GTA}$ | 51 | 0 | 87 | 110.29 |
| DisCup | 54 | 0 | 76 | 113.85 |
| DisCup$_{GTA}$ | 60 | 1 | 81 | 146.96 |

Table 41: Automatic evaluation result of the large-scale
LM for fear topic

| Method | Accuracy (↑) | Toxicity (↓) | Grammar (↑) | PPL (↓) |
|---|---|---|---|---|
| GPT-2 | 48 | 0 | 89 | 92.04 |
| GeDi | 59 | 0 | 80 | 64.19 |
| GeDi$_{GTA}$ | 55 | 1 | 90 | 91.83 |
| DisCup | 73 | 0 | 83 | 131.93 |
| DisCup$_{GTA}$ | 66 | 0 | 78 | 119.69 |

Table 42: Automatic evaluation result of the large-scale
LM for surprise topic

| Method | Accuracy (↑) | Toxicity (↓) | Grammar (↑) | PPL (↓) |
|---|---|---|---|---|
| GPT-2 | 38 | 0 | 76 | 131.57 |
| GeDi | 31 | 1 | 75 | 136.64 |
| GeDi$_{GTA}$ | 32 | 0 | 76 | 135.27 |
| DisCup | 35 | 0 | 67 | 100.52 |
| DisCup$_{GTA}$ | 35 | 0 | 71 | 117.87 |

Table 43: Automatic evaluation result of the large-scale
LM for business topic

| Method | Accuracy (↑) | Toxicity (↓) | Grammar (↑) | PPL (↓) |
|---|---|---|---|---|
| GPT-2 | 83 | 0 | 81 | 24.06 |
| GeDi | 86 | 0 | 82 | 22.4 |
| GeDi$_{GTA}$ | 87 | 0 | 84 | 23.32 |
| DisCup | 72 | 0 | 85 | 14.12 |
| DisCup$_{GTA}$ | 76 | 0 | 79 | 21.32 |

Table 44: Automatic evaluation result of the large-scale
LM for entertainment topic

| Method | Accuracy (↑) | Toxicity (↓) | Grammar (↑) | PPL (↓) |
|---|---|---|---|---|
| GPT-2 | 93 | 0 | 80 | 27.62 |
| GeDi | 91 | 0 | 79 | 23.75 |
| GeDi$_{GTA}$ | 95 | 0 | 78 | 25.53 |
| DisCup | 99 | 0 | 80 | 14.48 |
| DisCup$_{GTA}$ | 95 | 1 | 75 | 24.99 |

Table 45: Automatic evaluation result of the large-scale
LM for politics topic

| Method | Accuracy (↑) | Toxicity (↓) | Grammar (↑) | PPL (↓) |
|---|---|---|---|---|
| GPT-2 | 84 | 0 | 85 | 25.45 |
| GeDi | 89 | 0 | 83 | 27.35 |
| GeDi$_{GTA}$ | 86 | 0 | 83 | 24.02 |
| DisCup | 90 | 0 | 86 | 17.61 |
| DisCup$_{GTA}$ | 86 | 0 | 80 | 26.78 |

Table 46: Automatic evaluation result of the large-scale
LM for sport topic

| Method | Accuracy (↑) | Toxicity (↓) | Grammar (↑) | PPL (↓) |
|---|---|---|---|---|
| GPT-2 | 90 | 0 | 73 | 72.91 |
| GeDi | 86 | 1 | 70 | 75.44 |
| GeDi$_{GTA}$ | 90 | 0 | 69 | 77.94 |
| DisCup | 96 | 0 | 80 | 46.38 |
| DisCup$_{GTA}$ | 92 | 0 | 73 | 69.63 |

Table 47: Automatic evaluation result of the large-scale
LM for tech topic

| Method | Accuracy (↑) | Toxicity (↓) | Grammar (↑) | PPL (↓) |
|---|---|---|---|---|
| GPT-2 | 55 | 0 | 70 | 60.71 |
| GeDi | 61 | 0 | 74 | 62.28 |
| GeDi$_{GTA}$ | 53 | 0 | 74 | 59.16 |
| DisCup | 45 | 0 | 79 | 33.58 |
| DisCup$_{GTA}$ | 56 | 0 | 76 | 52.81 |