# OpenReview forum: "GTA: Gated Toxicity Avoidance for LM Performance Preservation"
_EMNLP/2023/Conference — EMNLP 2023 Findings_

### Official Review · Reviewer_Byj4 · 2023-08-03

**Soundness:** 3

**Excitement:**

2: Mediocre: This paper makes marginal contributions (vs non-contemporaneous work), so I would rather not see it in the conference.

**Paper Topic And Main Contributions:**

This paper focuses on the issue that existing detoxification methods suffer from quality degradation and inference overhead. To this end, the authors propose a simple model-agnostic gating method for reducing toxicity during language generation while preserving a language model's generation performance, such as topic consistency and grammar. The key is to apply a binary toxic classifier to help refine a toxic token to a nontoxic one. Extensive experiments on three datasets show that gated detoxifiers achieve comparable levels of toxic reduction while maintaining good generation performance.

**Questions For The Authors:**

A. Due to PPLM, GeDi, and DExperts being controllable generation models for detoxification, applying the gated detoxifier on top of them is easy to improve language quality (e.g., topic accuracy, grammar) while maintaining low toxicity. How about the performance for GPT-2 with the gated detoxifier?

**Reasons To Accept:**

1. The motivation of this paper is intuitive and easy to understand, with a solid literature review and necessary preliminaries. Besides, the proposed gated detoxifier is simple and easy to generalize.

2. Experiments are well-organized, with clear research questions (RQ1 to RQ6) to answer one by one. Experimental results show that gated detoxifiers achieve comparable levels of toxic reduction while maintaining good generation performance.

**Reasons To Reject:**

1. The main contributions of this paper are limited. The key to the proposed gated detoxifier is to apply a pre-trained binary toxic classifier to help refine a toxic token to a non-toxic one. It's hard to see key differences compared with existing controllable text generation methods for detoxification.

2. There is a lack of necessary technical details in the proposed method, e.g., how the toxic classifier is trained or obtained, how the gating mechanism works since the authors mentioned "gate threshold $\theta$" in Lines 431-433 but no introduction in Section 4.

3. Since the proposed method is very simple, I would expect it to be a simple yet effective approach. However, the performance gains of automatic evaluation are not remarkable enough, especially the Toxicity metric.

**Reproducibility:**

4: Could mostly reproduce the results, but there may be some variation because of sample variance or minor variations in their interpretation of the protocol or method.

**Reviewer Confidence:**

4: Quite sure. I tried to check the important points carefully. It's unlikely, though conceivable, that I missed something that should affect my ratings.

**Typos Grammar Style And Presentation Improvements:**

The overall writing needs to be carefully proofread. e.g., there is a lack of "." in Line 327 and Line 333.

---

> ### Author Rebuttal · Authors · 2023-08-29
>
> Thank you for your detailed and insightful review. Here are the comments on your review.
>
> 1 & 3) Main contributions:
>
> In order to commercialize the generative model, detoxification and subsequent performance degradation have received a lot of attention. However, it mainly focused on partial analysis - linguistic quality [1] or gender/regional/racial bias [2] -. To the best of our knowledge, this is the first analysis work of various aspects of LM performance degradation (topic preserving, grammar, and perplexity) of detoxification. We also conducted extensive experiments with four baselines (PPLM, GeDi, DExperts, and DisCup).
>
> We analyze the performance degradation of existing detoxification methods, and we also propose a simple gated detoxifier that controls the detoxifier to lower the toxicity without sacrificing LM performance. In addition, the gated detoxifier significantly decreases the cost of original detoxifiers of the Controllable Text Generation (CTG) technique. Even though GeDi and DExperts show low toxicity, they have not been used in practice because of their inference time cost. This is the first approach that can significantly reduce the overhead of CTG techniques.
>
> 2\) Technical details:
>
> Q. Due to PPLM, GeDi, and DExperts being controllable generation models for detoxification, applying the gated detoxifier on top of them is easy to improve language quality (e.g., topic accuracy, grammar) while maintaining low toxicity. How about the performance for GPT-2 with the gated detoxifier?
>
> We conducted an experiment which add a gate to GPT-2 without a detoxifier. When a toxic token is generated, sampling is performed again. This method had no performance difference from only using GPT-2. In the gated GPT-2, toxic tokens are continuously generated even if toxic tokens are resampled from one token to another because the context has a great influence on toxic text. Therefore, at that moment, the detoxifier must intervene and change the probability distribution.
>
> Thanks for highlighting the typographical errors. We'll ensure several rounds of proofreading are done before the final submission.
>
> [1] Gu, Yuxuan, et al. ``Improving controllable text generation with position-aware weighted decoding.`` Findings of the Association for Computational Linguistics: ACL 2022. 2022.
> [2] Xu, Albert, et al. ``Detoxifying language models risks marginalizing minority voices.`` arXiv preprint arXiv:2104.06390 (2021).

---

### Official Review · Reviewer_Gtqh · 2023-08-04

**Soundness:** 4

**Excitement:**

4: Strong: This paper deepens the understanding of some phenomenon or lowers the barriers to an existing research direction.

**Paper Topic And Main Contributions:**

Detoxifying methods seek to modify the model's generation to mitigate potentially toxic outputs. Usually, these methods are computationally expensive, both in terms of time and computation. To solve this issue, this work presents gated detoxifiers. Instead of applying the detoxifier for every single inference step,  it is only applied to steps scored above a certain threshold. This method allows a more efficient inference, as the detoxifier is only applied to steps that are considered potentially toxic.  Experimental results show that this method is able to keep the performance of previous detoxifiers while reducing performance degradations when using the models on downstream tasks.

**Reasons To Accept:**

- The paper is well-written and easy to follow.
- The proposed method is intuitive and easy to implement in a variety of detoxifiers.
- Experimental results show that the method is effective at retaining the model's performance.

**Reasons To Reject:**

- It would be interesting to see if the results are consistent when applied to other architectures (LLama, Falcon, GPT3, etc), even though the experimental section is quite thorough.

**Reproducibility:**

5: Could easily reproduce the results.

**Reviewer Confidence:**

4: Quite sure. I tried to check the important points carefully. It's unlikely, though conceivable, that I missed something that should affect my ratings.

---

> ### Author Rebuttal · Authors · 2023-08-29
>
> Thank you for your detailed and insightful review.
> Here are the comments on your review.
>
> 1) Modern LLM:
> We also validate our model on LLaMA (meta-llama/Llama-2-7b-chat-hf [1] in the huggingface). This is a instruction-tuned model with Reinforcement Learning from Human Feedback (RLHF). We have not tried applying gated detoxifier to modern LLM. We confirmed that toxicity is still a major issue in this model. We generated a total of 6000 short tweets which containing 6 emotions. Afterwards, we evaluated how accurate with emotion these tweets were and how toxic they were. For evaluation, the classifiers used in the paper were used.
>
> - anger: 57.7\% accuracy, 2.2\% toxicity
> - fear: 58.5\% accuracy, 1.1\% toxicity
> - joy: 88\% accuracy, 0.2\% toxicity
> - love: 13.3\% accuracy, 0.1\% toxicity
> - sadness: 64.5\% accuracy, 0.2\% toxicity
> - surprise: 20.5\% accuracy, 0.2\% toxicity
>
> The average toxicity is 0.67\%. It can be seen that llama trained with RLHF still shows high toxicity in certain topics. Compared to GeDi's 0.03\% and DExperts' 0.05\%, that's a huge toxicity difference. Even with modern LLM, we can see that there is still a lot of room for improvement.
>
> [1] Touvron, Hugo, et al. ``Llama 2: Open foundation and fine-tuned chat models.`` arXiv preprint arXiv:2307.09288 (2023).

---

### Official Review · Reviewer_9wix · 2023-08-13

**Soundness:** 3

**Excitement:**

3: Ambivalent: It has merits (e.g., it reports state-of-the-art results, the idea is nice), but there are key weaknesses (e.g., it describes incremental work), and it can significantly benefit from another round of revision. However, I won't object to accepting it if my co-reviewers champion it.

**Paper Topic And Main Contributions:**

This work tackles the problem of improving detoxification techniques which aim to prevent toxic content in language model generations. Authors propose a gating scheme that is pretty general and agnostic to the detoxification scheme used (weighted decoding, PPLM, GeDi, DExperts) using a toxicity classifier. The gating scheme decides whether or not at a specific token t (decoding time-step) intervention from the detoxification scheme is required. This is a pretty clever insight. Toxic content is rare, therefore, for a large majority of the tokens/decoding steps, we don't need the guiding detoxifiers to intervene.

The experimental results indicate that their gating mechanism can better preserve the linguistic quality of generated text from the base LM compared to the baseline detoxification schemes without any gating function. Another benefit achieved is in terms of decoding speed - with the gating scheme, the intervention of the detoxifiers/guiding models occurs for a pretty small fraction of tokens in the LM's generation, thus the resulting generation scheme is faster than using the detoxifier at each decoding step.

**Questions For The Authors:**

A - I could not find the architecture and training details of the gating function/model.

B - Line 480: Why is this the case? Why are toxicity classifiers less expensive than detoxifiers?

**Reasons To Accept:**

- Clever trick to improve detoxification schemes both in linguistic quality and speed of generation
- Detoxification of LM generation is an important problem to address
- Exhaustive experiments; the gating scheme is applied to several detoxifiers in many different settings
  - Research goal is studied through multiple research questions, each of which is empirically answered.

**Reasons To Reject:**

- Relevance of the approaches studied and improved in this work:
  - The proposed method improves detoxification schemes that were proposed after the release of the GPT-2 model. Toxicity in LM generation has received a lot of attention since then, and the approach that has eventually emerged prominent to prevent toxicity and improve LM helpfulness is fine-tuning (supervised) on good quality data and through reinforcement learning based on human feedback (RLHF). This is very different from the approach of detoxifiers being studied in this work. Any overhead on top of a base LM is not preferred during the inference stage of decoding. Thus, most practical LMs are being additionally fine-tuned based on human preferences (supervised FT and RLHF), while the approach of adopting a guiding detoxifier has not received much adoption in practice. Given this landscape, I'm concerned that the proposed approach, while being scientifically interesting, tries to improve detoxification techniques that are not highly relevant in contemporary research.
  - Many more language models have been released after GPT-2, which have not been studied in this work - FlanT5 (Scaling Instruction-Finetuned Language Models) and/or Llama. Evaluating the effectiveness of the proposed gating scheme on these recently proposed LMs would make the contribution more prominent.

- Significance of the results:
  - RQ1: GeDi seems to be at par with the base LM in terms of fluency as per the results in Table 1. The authors claim that GeDi hurts LM performance is therefore not convincing.
  - RQ2: While the gating scheme helps in improving the linguistic quality of PPLM and DExperts, GeDi doesn't seem to need the gating function for improving linguistic quality (Table 3).

**Reproducibility:**

4: Could mostly reproduce the results, but there may be some variation because of sample variance or minor variations in their interpretation of the protocol or method.

**Reviewer Confidence:**

5: Positive that my evaluation is correct. I read the paper very carefully and I am very familiar with related work.

**Typos Grammar Style And Presentation Improvements:**

- Title needs some rephrasing.
- Presentation clarity: The paper is hard to read and follow at several places. The writing style is extremely colloquial (eg. Line 027) and needs major refinement. Goal of the paper and how the proposed scheme achieves it is not clearly communicated.
- For a reader who's not familiar with the literature on detoxification techniques, the proposed approach is hard to follow in the current draft. Introduction doesn't explain the method clearly enough

Line 023: toxic --> toxicity

Line 044, Line 050 --> citations for these claims

Line 095: what does toxic degeneration mean?

Lines 180-200 (Section 3.1, partially Sec 3.2) - does not seem necessary after the related work description

Line 232: proof? qualitative or quantitative?

Line 235: writing style is too colloquial,

Line 456-450: What is %p?

Line 480: should it be instead called 'toxicity classifier'?

Line 400-405: unclear what the authors are trying to communicate.

---

> ### Author Rebuttal · Authors · 2023-08-29
>
> Thank you for your detailed and insightful review.
> Here are the comments on your review.
>
> 1\) Instruction-tuned & 2) Reinforcement Learning from Human Feedback (RLHF):
>
> To focus on the toxicity in the paper, we select CTG methods for the baselines. The objectives of instruction-tuned and RLHF are toxicity (harmfulness) and helpfulness. Thank you for your constructive comments, and we validate toxicity on LLaMA (meta-llama/Llama-2-7b-chat-hf [1] in the huggingface). This is a instruction-tuned model with Reinforcement Learning from Human Feedback (RLHF). We utilized the following prompts to generate 6,000 tweets containing six different emotions. The emotion and toxic classifiers are same as used in the paper.
>
> We also observed that CTG methods show the better performance on toxicity. The average toxicity of LLaMA is 0.67\% (anger: 2.20\%, fear: 1.10\%, joy: 0.20\%, love: 0.10\%, sadness: 0.20\%, and surprise: 0.20\%). When using the CTG method, we observed that GeDi yielded a remarkably low toxicity rate of 0.03\% on the GPT-2 (125M) model, while DExperts exhibited a similar low toxicity rate of 0.05\%. Given the substantial reduction in toxicity, the CTG technique proves to be highly valuable as a means to mitigate toxicity. We will supplement that modern RLHF models like LLaMA also encounter similar challenges.
>
>
> 3\) Any overhead on top of a base LM is not preferred during the inference stage of decoding.
>
> It is true that the CTG technique has performance overhead. This overhead was concerned significantly when deploying real-world service. In the light of these concerns, our methods decrease such overhead. The generation of toxic text poses substantial risks in practical service scenarios. Consequently, our work holds the potential to be applied in contexts where the generation of toxic content carries significant risks.
>
>
> 4\) Significance of the results(RQ1 & RQ2: GeDi performance):
>
> RQ1: GeDi seems to be at par with the base LM in terms of fluency as per the results in Table 1. The authors claim that GeDi hurts LM performance is therefore not convincing.
>
> RQ2: While the gating scheme helps in improving the linguistic quality of PPLM and DExperts, GeDi doesn't seem to need the gating function for improving linguistic quality (Table 3).
>
> Answer:
> On average, GeDi appears to be rather better on linguistic quality metrics. However, when looked at by topic, the results are different. Detoxifier's performance degradation occurs differently depending on the topic and metric. This table is the evaluation result of GPT-2, GeDi, and GeDi\_{gated} at small scale.
>
>
> These are changes of GeDi's linguistic quality from GPT-2
> - Grammar in emotion topic group: 88.31 -> 91.06 (+2.85)
> - Grammar in sentiment topic group: 88.22 -> 82.09 (-6.13)
>
> GeDi increases linguistic quality only in the emotion topics. The reason is that the texts in the emotion dataset are missing grammatical elements such as comma and apostrophe. On the other hand, in the sentiment dataset, the linguistic quality is rather greatly reduced. Sentiment has 2 topics, emotion has 6 topics. As a result, when micro-averaged, the difference in emotion is more prominent. We will add these qualitative results in the appendix.
>
>
>
> #### Questions:
>
> Q.A) I could not find the architecture and training details of the gating function/model.
>
> We will add the details of architecture and training information. We used the s-nlp/roberta\_toxicity\_classifier model, publicly available on Hugging Face, as our gating model. This model is trained for toxicity classification task. This model is fine-tuned RoBERTa[2] model for toxicity classification task. The dataset used for training are Jigsaw datasets ([Jigsaw 2018](https://www.kaggle.com/c/jigsaw-toxic-comment-classification-challenge), [Jigsaw 2019](https://www.kaggle.com/c/jigsaw-unintended-bias-in-toxicity-classification), [Jigsaw 2020](https://www.kaggle.com/c/jigsaw-multilingual-toxic-comment-classification)), containing around 2 million examples. The classifiers perform closely on the test set of the first Jigsaw competition, reaching the AUC-ROC of 0.98 and F1-score of 0.76. Since this is not the model we trained, the training details are unknown.
>
>
>
> Q.B) Line 480: Why is this the case? Why are toxicity classifiers less expensive than detoxifiers?
>
> Our gated detoxifier is less expensive than the original detoxifiers. The total parameters of the gate model (gating network + toxicity classifiers) are 110M, but GeDi requires 355M parameters and DExperts requires 1550M parameters (it uses two 775M models). In addition, the probability of generating a toxic sentence in gpt-2 is only 1.85\%. Even within a sentence, few tokens are toxic. Therefore, the generation speed can be reduced compared to applying a detoxifier to every generation step.
>
> To clarify our method, we will change our paper title for the camera-ready version.
>
> Thank you for pointing out typo errors. We will conduct multiple proof-reading before submitting the final version.
>
>
> [1] Touvron, Hugo, et al. ``Llama 2: Open foundation and fine-tuned chat models.`` arXiv preprint arXiv:2307.09288 (2023).
>
> [2] Liu, Yinhan, et al. ``Roberta: A robustly optimized bert pretraining approach.`` arXiv preprint arXiv:1907.11692 (2019).

---

### Official Review · Reviewer_4a3Y · 2023-08-14

**Typos Grammar Style And Presentation Improvements:** 1. Please consider clarifying what yo…
**Soundness:** 4

**Excitement:**

3: Ambivalent: It has merits (e.g., it reports state-of-the-art results, the idea is nice), but there are key weaknesses (e.g., it describes incremental work), and it can significantly benefit from another round of revision. However, I won't object to accepting it if my co-reviewers champion it.

**Paper Topic And Main Contributions:**

This paper explored the limitations of the existing detoxifiers, in terms of topic consistency, grammar, and fluency. To alleviate the issue the authors proposed a model-agnostic solution called the Gated Detoxifier. The authors show that the proposed method can successfully preserves the model's performance while achieving comparable toxicity reduction levels.

**Questions For The Authors:**

1. Is there any particular reason that you did not include the SOTA detoxifier, ParaDetox?

**Reasons To Accept:**

1. This work explores the idea of improving the text quality of the detoxifiers, which is a motivating research topic.
2. The authors propose a gated detoxifier fused with guided decoding method to preserve the text quality while reducing the toxicity.
3. Different evaluation process shows the improvement of text quality in terms of topic preservation, grammar, and fluency.

**Reasons To Reject:**

1. The novelty of this work is bit limited. This works seems to incorporate guided generation with controllable non toxic text generation. It would be encouraging to see the model's improvement in accuracy, grammar, and toxicity metrics. But the evaluation of different baselines shows that the proposed method sometimes sacrifices on toxicity metric (Table 3, 5, 6) to generate more topic-preserving text.
2. The topmost priority of this line of work should be reducing toxicity. However, the experimental results are not convincing enough to conclude that the proposed method is reliable to reduce the toxicity.
3. The authors use the term 'detoxifier', which is a bit confusing, because prior works also use the term 'detoxification' for toxic-to-non toxic style transfer [1]. This line of research is different from open-domain non toxic text generation. Although Figure 1 gives the sense that this is a style transfer work (preserve the meaning of the source text), the evaluation strategy and the rest of the examples give the opposite idea. Please clarify on this.

References:

[1] Logacheva et al., ParaDetox: Detoxification with Parallel Data, ACL 2022.

**Reproducibility:**

3: Could reproduce the results with some difficulty. The settings of parameters are underspecified or subjectively determined; the training/evaluation data are not widely available.

**Reviewer Confidence:**

4: Quite sure. I tried to check the important points carefully. It's unlikely, though conceivable, that I missed something that should affect my ratings.

---

> ### Author Rebuttal · Authors · 2023-08-29
>
> Thank you for your detailed and insightful review.
> Here are the comments on your review.
>
>
> 1 & 2) Novelty:
>
> The novelty of our work is an analysis of existing detoxifiers - detoxifiers decrease toxicity and also decrease LM performance - and our gated detoxifier can resolve the problems. With the marginal toxicity sacrifice (0.04 in PPLM, 0.00 in GeDi and DExperts), the gated detoxifier gains LM generation performance (topic preserving, grammar, and perplexity).
>
>
> 3\) Confusing term `detoxifier`:
>
> Our term `detoxifier` is used for `guided decoder` of LM for non-toxic text generation. We picked a detoxifier from the previous detoxification studies [1,2,3], but we agree that the term is confusing. We will clarify our task and change `detoxifier` to `guided decoder` for the camera-ready version.
>
>
>
> Q. Is there any particular reason that you did not include the SOTA detoxifier, ParaDetox?
>
> A. The proposed gated detoxifier is the guided decoder which can be applied to the CTG method, whereas its applicability to the text style transfer task is limited. Instead, we selected the state-of-the-art CTG method, DisCup[3], proposed in 2022. However, through a straightforward experiment, we also observed ParaDetox encountering challenges in text style transfer. Leveraging a toxic classifier, we curated a subset of 628 toxic samples from the emotion dataset. Then we attempted to transform these samples into non-toxic using ParaDetox. We evaluate the detoxified texts using the emotion classifier. We were able to observe variations in topic accuracy following detoxification, depending on the topic. The performance degradation (avg. -13.49\%) also occurred in the ParaDetox. We used emotion and toxicity classifier which also used in the paper.
>
> The obtained results are as follows:
> - anger: 100.0 -> 57.17\% accuracy, 146 samples
> - fear: 100.0 -> 84.85\% accuracy, 56 samples
> - joy: 100.0 -> 96.36\% accuracy, 106 samples
> - love: 100.0 -> 90.91\% accuracy, 50 samples
> - sadness: 100.0 -> 89.79\% accuracy, 255 samples
> - surprise: 100.0 -> 100\% accuracy, 16 samples
>
>
> Thank you for pointing out typo errors. We will conduct multiple proof-reading before submitting the final version.
>
> [1] Krause, Ben, et al. ``Gedi: Generative discriminator guided sequence generation.`` arXiv preprint arXiv:2009.06367 (2020).
>
> [2] Gu, Yuxuan, et al. ``Improving controllable text generation with position-aware weighted decoding.`` Findings of the Association for Computational Linguistics: ACL 2022. 2022.
>
> [3] Zhang, Hanqing, and Dawei Song. ``DisCup: Discriminator cooperative unlikelihood prompt-tuning for controllable text generation.`` arXiv preprint arXiv:2210.09551 (2022).

---

### Meta-Review · Area_Chair_zyXr · 2023-09-17

**Recommendation:** 3

**Metareview:**

The paper introduces the concept of gated detoxifiers as a solution to improve the text quality of the existing guided decoder of LM for non-toxic text generation. The proposed solution is designed to be model-agnostic and aims to preserve the linguistic quality of the text while reducing toxicity.

Pros:
1. Addresses a significant problem of improving text quality in non-toxic text generation.
2. The proposed method is model-agnostic.
3. Positive experimental results in retaining model performance and reducing computational overhead.

Cons:
1. Limited novelty with similarities to existing methods.
2. Lack of clarity. Two reviewers point out that the term "detoxifier" is confused. It doesn't reflect the main purpose of the paper. Reviewers also mention that the technical and implementation details were missed.
3. Writing and presentation need refinement. Reviewers 1, 2 and 4 point out many language errors.

---

### Decision · Program_Chairs · 2023-10-07

**Decision:**

Accept-Findings

**Comment:**

The paper introduces the concept of gated detoxifiers as a solution to improve the text quality of the existing guided decoder of LM for non-toxic text generation. The proposed solution is designed to be model-agnostic and aims to preserve the linguistic quality of the text while reducing toxicity.

Pros:
1. Addresses a significant problem of improving text quality in non-toxic text generation.
2. The proposed method is model-agnostic.
3. Positive experimental results in retaining model performance and reducing computational overhead.

Cons:
1. Limited novelty with similarities to existing methods.
2. Lack of clarity. Two reviewers point out that the term "detoxifier" is confused. It doesn't reflect the main purpose of the paper. Reviewers also mention that the technical and implementation details were missed.
3. Writing and presentation need refinement. Reviewers 1, 2 and 4 point out many language errors.